# Therapeutic Potential of PI3K/AKT/mTOR Pathway in Gastrointestinal Stromal Tumors: Rationale and Progress

**DOI:** 10.3390/cancers12102972

**Published:** 2020-10-14

**Authors:** Yi Duan, Johannes Haybaeck, Zhihui Yang

**Affiliations:** 1Department of Pathology, the Affiliated Hospital of Southwest Medical University, Luzhou 646000, China; duanyi018@swmu.edu.cn; 2Department of Pathology, Neuropathology and Molecular Pathology, Medical University of Innsbruck, 6020 Innsbruck, Austria; 3Diagnostic & Research Center for Molecular BioMedicine, Institute of Pathology, Medical University of Graz, 8010 Graz, Austria

**Keywords:** gastrointestinal stromal tumors, PI3K/AKT/mTOR, inhibitor, eIFs

## Abstract

**Simple Summary:**

Most gastrointestinal stromal tumors (GISTs) arise due to gain-of-function mutations of *KIT* and *PDGFRA*, encoding the receptor tyrosine kinase (RTK). The introduction of the RTK inhibitor imatinib has significantly improved the management of GISTs; however, drug resistance remains a challenge. Constitutive autophosphorylation of RTKs is associated with the activation of the PI3K/AKT/mTOR pathway. Especially, this pathway plays a pivotal role in mRNA translation initiation, directly regulated by eukaryotic initiation factors (eIFs). This review highlights the progress for targeting PI3K/AKT/mTOR-dependent mechanisms in GISTs and explores the relationship between mTOR downstream eIFs and the development of GISTs, which may be a promising future therapeutic target for this tumor entity.

**Abstract:**

Gastrointestinal stromal tumor (GIST) originates from interstitial cells of Cajal (ICCs) in the myenteric plexus of the gastrointestinal tract. Most GISTs arise due to mutations of *KIT* and *PDGFRA* gene activation, encoding the receptor tyrosine kinase (RTK). The clinical use of the RTK inhibitor imatinib has significantly improved the management of GIST patients; however, imatinib resistance remains a challenge. The phosphatidylinositol 3-kinase (PI3K)/protein kinase B (AKT)/mammalian target of rapamycin (mTOR) pathway is a critical survival pathway for cell proliferation, apoptosis, autophagy and translation in neoplasms. Constitutive autophosphorylation of RTKs has an impact on the activation of the PI3K/AKT/mTOR pathway. In several preclinical and early-stage clinical trials PI3K/AKT/mTOR signaling inhibition has been considered as a promising targeted therapy strategy for GISTs. Various inhibitory drugs targeting different parts of the PI3K/AKT/mTOR pathway are currently being investigated in phase Ι and phase ΙΙ clinical trials. This review highlights the progress for PI3K/AKT/mTOR-dependent mechanisms in GISTs, and explores the relationship between mTOR downstream signals, in particular, eukaryotic initiation factors (eIFs) and the development of GISTs, which may be instrumental for identifying novel therapeutic targets.

## 1. Introduction

### 1.1. Clinical Features

Gastrointestinal stromal tumor (GIST) is the most prevalent mesenchymal tumor in the gastrointestinal (GI) tract. Generally known, this tumor entity arises from the intestinal cells of Cajal (ICCs), a line similar to GI pacemaker cells, or multifunctional stem cells differentiating into ICCs [1]. Søreide and colleagues reviewed 29 studies of more than 13,550 GIST patients from 19 countries. The median age was mid 60 s across most studies. The incidence rate was 10 to 15 per million per year. Hong Kong, Shanghai, Taiwan and Norway (Northern part) were the highest incidence areas, with up to 19–22 per million per year; notably, the Shanxi province of China was the lowest incidence area, with 4.3 per million per year [2]. GISTs occur throughout the GI tract, most commonly in the stomach (60–70%), small intestine (20–25%), and less frequently in the colon and rectum (5%) [3,4]. Clinical manifestations of GISTs are highly diverse according to their size and location [5]. Most frequent symptoms are GI bleeding, abdominal pain and mass-related symptoms. Some patients occasionally present with an asymptomatic tumor [5,6]. In rare cases, patients present with symptoms such as the Carney triad, which is associated with GISTs, pulmonary chondroma [6], and paraganglioma [7], or neurofibromatosis type 1 [8]. GISTs are usually discovered during surgery, endoscopy or imaging studies.

Computer tomography (CT) is the most common imaging examination procedure for diagnosing GISTs prior to surgery [9]. GIST imaging features are usually present as dense mass with a clear boundary, whereas some of them are accompanied by cystic degeneration and central necrosis [10,11]. Furthermore, the CT scan is of great significance for the diagnosis of metastases of malignant GISTs. With CT scanning, for malignant GIST, hepatic or peritoneal metastases are most common, and lymph node metastases are rare [10,12]. In the consensus report of the German GIST Imaging Working Group, contrast-enhanced CT was defined as the preferred method for GIST imaging [13]. At present, the post-surgical molecularly targeted therapy is the best strategy for advanced GIST patients, leading to significantly improved overall survival. GIST patients should undergo a CT scan in order to evaluate the response to the molecularly targeted therapy [11,13]. Another method for assessing therapy response is the use of magnetic resonance imaging (MRI), which is based on the evaluation of criteria similar to contrast-enhanced CT. However, MRI may be less reliable due to its difficulty in reproducible measurements of the signal intensity (SI) change, and thereby cannot be considered as the primary standard for therapeutic response assessment in GIST patients [9]. MRI is only suggested for those GIST patients with a liver-related problem (e.g., potential hepatic resection) or contraindications to CT [13]. Positron emission tomography (PET)/CT is used for assessing an early response or for interpreting uncertain results obtained by morphological imaging. Furthermore, it is helpful for predicting the malignant potential of GISTs as far as sensitivity and specificity is concerned [14]. The final diagnosis of GISTs is always based on morphology and immunohistochemistry.

In gross patterns, the features of GISTs are diverse. Miettinen and colleagues showed that sizes of GISTs range from 0.3 cm to 44 cm, with a median size of 6.0 cm [15]. Usually, the shape of the tumor is described as nodular, lobulated or bosselated with well circumscribed borders [7,15]. Histopathologically, GISTs are mainly divided into three pathological types, including spindle cell type, epithelioid cell type and mixed, spindle-epithelioid type. Here, the spindle cell type most commonly accounts for a proportion of 70%, and there is no significant numeric difference between epithelioid cell type (a proportion of 17%) and mixed type (a proportion of 13%) (Table 1) [16]. 

The pathological feature of spindle cell type is a spiral or palisading arrangement, in which the nuclei of tumor cells are fat to slender or fusiform with perinuclear vacuolization, and the cytoplasm of the tumor cells is eosinophilic or light-stained with mild atypia (Figure 1A). Different degrees of mitotic activity may be seen. The epithelioid cell type is mainly composed of cords or small nests of epithelioid cells with abundant eosinophilic cytoplasm, without distinctive cell borders (Figure 1B). The mixed type is composed of spindly and epithelioid cells with a staggered arrangement (Figure 1C). Until recently, KIT (CD117), a tyrosine kinase receptor, has been the main tool for the validation of GISTs with a positivity rate in immunohistochemistry of more than 90% (Table 1) [3,15,17]. Most GISTs are strongly positive for CD117 (Figure 1D), 10% of them are weakly or focally positive, and less than 5% of them may be negative [3,18]. In the last 10 years, DOG1 has been increasingly studied (Figure 1E) [19,20]. It is a calcium-dependent receptor-activated chloride ion channel protein, and provides a good sensitivity and specificity in the diagnosis of GISTs. Miettinenet and colleagues reported DOG1 immunostaining as positive in 96% (Table 1) [21]. Currently, the combination of CD117 and DOG1 is recommended to assist in the diagnosis of GISTs. In the few cases where either CD117 or DOG1 are negative, a molecular analysis should be considered. 

### 1.2. Molecular Pathogenesis

*c-KIT* mutations occur in approximately 70–85% of GISTs. *c-KIT* gene encodes a type III RTK that is activated when combined with a ligand called steel factor or stem cell factor (SCF) [22,23]. Later, pharmacologic findings have proven that the *c-KIT* is of paramount importance for the selection of therapeutic targets. In GISTs, *c-KIT* mutations are most common in exon 11, accounting for 90% (Table 2). These GISTs are most sensitive to the tyrosine kinase inhibitor imatinib [22,23]. The *KIT* exon 11 mutation is subdivided into three mutation sub-types, whereas 40% of exon 11 mutations are deletions, followed by substitutions (30%) and duplication (16%). Most exon 11 deletion mutations involve codon 557 and/or 558, leading to Trp557_Lys558 deletion [24]. Other *c-KIT* mutations occur in exon 9 (8%), in exon 13 (1%) or in exon 17 (1%) (Table 2), while exon 9 mutation mainly leads to Ala502_Tyr503 duplication, exon 13 mutation to Lys642Glu (K642E), and exon 17 mutation to Asn822Lys (N822K) [25,26].

Platelet-derived growth factor receptor-α gene (*PDGFRA*) is mutated in probably 5–8% of GISTs (Table 2), which encodes a type III RTK with PDGFA as its ligand [27]. *PDGFRA*-mutated GIST patients usually benefit from imatinib or other TKIs. *PDGFRA* mutations most commonly occur in exon 18 but rarely in exons 12 and 14. ASP824Val (D842V) substitution is considered as a resistance factor to imatinib and is most commonly found in *PDGFRA* exon 18 mutation, whereas other subtypes of *PDGFRA* mutations are sensitive to imatinib [28,29]. 

In addition, wild-type (wt) GISTs that lack *c-KIT* or *PDGFRA* mutations account for about 10–15% of GISTs (Table 2). They are subdivided into two subgroups: the succinate dehydrogenase (*SDH*)-deficient GIST and the non-*SDH* deficient GIST. *SDH*-deficiency leads to increased whole genome methylation in tumors, which is closely associated with Carney and wt GISTs [4,30]. In the non-*SDH* deficient GISTs, *BRAF*-mutated GISTs, NF1-related GISTs, *K/N-RAS* mutation-associated GISTs, and quadruple wt GISTs are included [31].

### 1.3. Prognostic Factors and Risk Stratification

Imatinib, targeting *KIT* and *PDGFRA*, has become the standard first-line adjuvant drug for GISTs, thus improving the prognosis of intermediate-high risk GIST patients [31,32]. For these GIST patients, imatinib can improve the 3-year recurrence-free survival rate [33]. Therefore, the National Institutes of Health (NIH) risk classification criteria are mainly used to evaluate imatinib adjuvant treatment. Moreover, the poor prognosis of GIST patients is commonly associated with large tumor volume, tumor ulceration, necrosis, high mitotic activity, tumor location outside the GI tract, infiltration and metastasis [34,35]. The biological behavior of GISTs can considerably differ, so better methods for accurate prediction of the clinical course are required. 

Although GIST patients have obviously benefitted from imatinib, with longer survival than those without treatment [34,36,37], drug resistance may be a serious problem. Primary resistance to imatinib in GISTs is closely related with gene mutation types [38]. Approximately 80% of GISTs harbor *c-KIT* mutations and 10% of them show a mutation in *PDGFRA* [22]. *c-KIT*/*PDGFRA* mutations influence the prognosis of GIST patients, and those molecular biomarkers can provide guidance for adjuvant chemotherapy, resulting in a better prognosis for GIST patients. Compared with other mutation types, *KIT* exon 11 mutations are related to a better response to imatinib and long-term survival [39]. *KIT* exon 11, involving codon 557/558 deletions, are associated with worse prognosis [40]. Conversely, *PDGFRA*-mutated GISTs are more prone to a benign behavior than those with codon 557/558 deletions [40,41]. Patients with a *PDGFRA-D824V* mutation are primarily resistant to imatinib [42,43]. Interestingly, even without adjuvant therapy with imatinib, *D842V*-mutated GISTs have a high recurrence-free survival rate, which suggests that *D842V* mutations may bear a low risk of recurrence and may thus predict a relatively good prognosis.

## 2. Overview of the PI3K/AKT/mTOR Pathway

Phosphatidylinositol 3-kinase (PI3K)/protein kinase B (AKT)/mammalian target of rapamycin (mTOR) pathway is necessary in many neoplasms. It has important effects on the basic intracellular functions of cell growth, apoptosis, translation, and cell metabolism [44,45]. It is not only vital in the process of carcinogenesis, but also important for identifying potential novel therapeutic targets. 

PI3Ks are lipid kinases that phosphorylate the inositol ring 3-hydroxyl group in inositol phospholipid and hence produce phosphatidylinositol (3,4,5)-trisphosphate (PIP3) [46]. They are key targets for signaling integration and downstream component activation. PI3Ks are considered to have three sub-classes named class Ι, II and III [47]. Further including class IA and class IB, class Ι PI3Ks are heterodimeric proteins composed of catalytic subunits and regulatory subunits. The former is activated by RTKs and the latter is activated by G protein-coupled receptors (GPCRs) [45,47]. *c-KIT* and *PDGFR* are members of the RTK family. When the ligand binds to its specific RTK, the signal transduction adaptor protein (such as insulin receptor substrate 1, IRS1) is phosphorylated, thereby leading to PI3K activation [48]. Class ΙA PI3Ks are widely present in carcinogenic processes. They consist of three catalytic subunit isoforms (p110α, p110β and p110δ) and five adaptor/regulatory subunit isoforms (p85α, p55α, p50α, p85β and p55γ). Three highly homologous p110 catalytic subunit isoforms are encoded by the respective p110-related genes (*PIK3CA*, *PIK3CB* and *PIK3CD*). P110α and p110β can promote angiogenesis and cell proliferation, whereas p110δ contributes to immune function [49]. Five p85 isoforms are encoded by the p85-related genes (*PIK3R1*, *PIK3R2* and *PIK3R3*), which are referred collectively as the consequence of splice variants. Class IB PI3Ks include the catalytic subunit p110γ and regulatory subunit isoforms (p84/p87 or p101) [45]. The gene *PIK3CG* encodes a p110 isoform (p110γ) that plays an important role in inflammation. Regulatory partners of p110γ do not suppress kinase activity, but act to promote activating signals. Genes *PIK3R6* and *PIK3R5* encode two regulatory subunit isoforms (p84/p87, p101), respectively [50]. Class II PI3Ks contain only one catalytic subunit analogue to p110, in which three isoforms (C2α, C2β and C2γ) are encoded by distinct genes (*PIK3C2A*, *PIK3C2B* and *PIK3C2G*), and not by adaptor/regulatory subunits; it regulates membrane trafficking and receptor internalization [51]. Class III PI3Ks are composed of a *PIK3C3*-encoded catalytic subunit (Vps34) and a *PIK3R4*-encoded regulatory subunit (Vps15), in which the function is associated with the regulation of autophagy [52]. 

At present, research on PI3K signal transduction is mainly focused on class Ι PI3Ks. By RTKs, activated class Ι PI3Ks convert the substrate phosphatidylinositol 4, 5-bisphosphate (PIP2) into PIP3 that fails to be cleaved by phospholipase Cγ1 [53]. In addition, phosphatase and tensin homolog (PTEN) cause the dephosphorylation of the third position of the PIP3 inositol ring, thereby transforming into PIP2 (Figure 2) [46,54]. Notably, PIP3 causes the recruitment of AKT and phosphoinositide-dependent protein kinase-1 (PDK1), allowing the latter to reach the plasma membrane through pleckstrin homology domains (PH domains) [55]. Then, class Ι PI3Ks activate a wide range of downstream effectors by directly binding PH domains. AKT is considered as a central mediator of the PI3K pathway, as well as a principal target of PIP3 [56]. It is the main kinase that consists of three isoforms—AKT1, AKT2 and AKT3—which are encoded by the *PKB*-related genes. In the N-terminal region of each isoform, the PH domain is composed of approximately 100 amino acids [57]. AKT directly integrates upstream signals from PI3Ks and the mTOR complex 2 (mTORC2) into mTOR complex 1 (mTORC1). Activation of AKT depends on phosphorylating serine473 (Ser473) by mTORC2 and threonine 308 (Thr308) by PDK1, respectively (Figure 2). Activated AKT inhibits Bcl2-related death protein (BAD), causing Bcl-2 to dissociate on the mitochondrial membrane, and ultimately inhibits cell apoptosis [47]. Subsequently, AKT relays signal following mTOR activation via suppressing tuberous sclerosis complex 1 and 2 (TSC1/TSC2), which is known to regulate cellular growth, protein translation, and autophagy [54,55]. Activated TSC1/TSC2, acting as a GTPase-activating protein (GAP) for the Ras homolog enriched in brain (Rheb), leads to inhibit the action of mTORC1 [47,58,59]. The TSC1/TSC 2 complex also transmits more signals to mTORC1. Extracellular signal-regulated kinase (ERK) is an effector kinase downstream of RAS that directly phosphorylates TSC2 at S664/540, activating mTORC1 by inhibiting the TSC1/TSC2 complex (Figure 2) [60,61]. The RAS pathway is activated when combined with GTP, thereby regulating cell growth. So, the RAS-ERK pathway regulates mTORC1 passing the TSC1/TSC2 complex [62]. Besides, TSC2 can be phosphorylated through other pathways. Glycogen synthase kinase 3 (GSK3) can phosphorylate and activate TSC2, thereby inhibiting mTORC1 signal. Furthermore, AMPK, a hypoxia-induced tumor suppressor, also inhibits the mTORC1 signal by activating the TSC1/TSC 2 complex when cells lack enough energy [63].

Activated mTORC1 phosphorylates its downstream effectors, such as ribosomal protein S6 kinase (S6K) and eukaryotic translation initiation factor 4E-binding protein 1 (4E-BP1) (Figure 2). On the one hand, activated S6K phosphorylates ribosomal protein S6 (rpS6), which in turn stimulates translation again. On the other hand, inactivated 4E-BP1 enhances the release of eukaryotic translation initiation factor 4E (eIF4E), considering 4E-BP1 as an inhibitor mediator in the initiation of translation [64,65]. Of note, 4E-BP1 plays a significant mechanistic role in oncogenesis, such as promoting cell growth and protein translation, and is also closely related to drug resistance [66,67]. Thus, mTORC1 is mainly involved in regulating cell growth, proliferation, and consequently controlling angiogenesis [59]. Activated mTORC1 also activates Hypoxia-inducible factor 1-α (HIF1-α) [68]. While, under normoxic conditions, the hydroxylation of HIF1-α is recognized by von Hippel Lindau protein (VHL), promoting polyubiquitinylation and proteasomal degradation of HIF1-α under hypoxic conditions, the binding of unhydroxylated HIF1-α and vascular endothelial growth factor (VEGF) enhances transcription and promotes angiogenesis [69,70]. Activated mTORC1 inhibits autophagy by suppressing the autophagy-specific gene 1 (Atg1) kinase complex that is required to stimulate autophagy. Atg1 triggers a negative feedback regulation on mTOR [71,72]. S6K, a downstream target of mTORC1, has been shown to contribute to the stimulation of the process of autophagy. Some studies have shown that Atg1 inhibits cell growth by down-regulating S6K [73]. 

It is well known that mTORC2 can directly respond to growth factors, such as insulin, through binding to its ribosomes. The PI3K pathway directly controls the activation of mTORC2. Activated S6K can inhibit the function of mTORC2 via a negative feedback loop (Figure 2) [55]. In addition, Sin1 causes the mTORC2 complex to recruit AKT and SGK, thus retaining the stability of the complex. Activated mTORC2 directly activates AKT via phosphorylation at Ser473 and SGK1 (S422), controlling ion migration and cell growth [74]. However, the upstream activators and downstream regulators of the mTORC2 pathway are poorly understood, unlike mTORC1.

## 3. Influence of the PI3K/AKT/mTOR Pathway on Proliferation, Apoptosis, Autophagy and Progression of GISTs

Activated RTKs, such as c-KIT and PDGFRA, activate multiple intracellular signaling pathways, including PI3K/AKT/mTOR, Ras/RAF/ERK and JAK/STAT signaling. Among these pathways, the PI3K/AKT/mTOR pathway is essential in GISTs—for example, for cell proliferation, apoptosis, autophagy, differentiation—and consequently is linked to chemotherapy resistance. The pathway hyperactivated by c-KIT auto-phosphorylation induces the development of GISTs [75,76]. Thus, the pathway may be a critical target in chemotherapy and a promising therapeutic strategy in GISTs.

### 3.1. Influence on Cell Proliferation

It is well established that mTOR can assist the integration of nutrient and mitogen signals into intracellular, thereby regulating cell growth and cell division; meanwhile, rapamycin, an mTOR inhibitor, inhibits cell growth and cell cycle [77]. The overexpression of constitutively active mutated S6K1 or wt eIF4E accelerated G1-phase progression, suggesting that S6K1 and 4E-BP1/eIF4E as downstream signals of mTOR, regulates cell proliferation to a certain extent by controlling the cell cycle [78]. In a study of 108 GIST patients, the phosphorylation of p70S6K and 4E-BP1 was separately detected in *KIT*-mutated GISTs at 38%, in *PDGFRA*-mutated GISTs at 83%, and in wt GISTs at 74% [79]. Interestingly, mTOR was overexpressed in *PDGFRA*-mutated and wt GISTs, indicating that mTOR-related inhibitors may have therapeutic effects on primary imatinib-resistant GISTs [79]. Pang and colleagues demonstrated that mTORC1 signaling was inactivated by DEPDC5, with the suppression of the phosphorylation of p70S6K and S6, resulting in reduced cell proliferation and subsequently cell-cycle arrest in GIST cells [80]. Li and colleagues showed that PI3K/AKT/mTOR signaling was inactivated due to *FANS* knockdown, especially with the attenuation of the activation of RPS6 and 4E-BP1, leading to the inhibition of proliferation and migration in GIST cells [81]. 

### 3.2. Influence on Apoptosis

As the first study linking apoptosis to prognosis of GIST patients, Wang and colleagues found in 2007 that the apoptotic index was gradually decreased in tumor tissue specimens of patients with GISTs. The authors suggested that programmed cell death may be avoided in the pathogenesis of GISTs [82]. More recently, PI3K/AKT/mTOR-regulated apoptosis has been widely investigated in GISTs. Activated PI3K can directly inhibit apoptosis in tumors [65]. AKT, an anti-apoptotic factor, mediates these PI3K-dependent cell survival responses [64]. In GISTs, the activation mutation of the upstream c-KIT and PDGFRA is the initial mechanism of the PI3K/AKT/mTOR signal activation [83]. Ma and colleagues demonstrated KIT expression as positively correlating with the cell proliferation marker Ki-67 and a converse correlation with the pro-apoptotic protein APAF. Thereby *c*-*KIT* participates in GIST tumorigenesis by inducing proliferation and reducing apoptosis [84]. Ihle and colleagues reported that the downregulation of miR-221 and miR-222 induced apoptosis via the KIT/AKT pathway in GIST cells; the KIT/AKT pathway has effects on tumor progression, controlling cell proliferation and apoptosis [85]. 

### 3.3. Influence on Autophagy

Autophagy is a highly conserved catabolic process that participates in cell survival and in preserving cell metabolic balance [71,73]. In various tumors, autophagy is critically controlled by the PI3K/AKT/mTOR pathway [86]. Via this pathway, increased expression levels of autophagy related genes may enhance metastatic spread in paranasal squamous cell carcinomas [87]. Autophagy can promote cell adaption and survival, but it also leads to cell death under particular conditions. Autophagy is a double-edged sword for drug resistance, as growing evidence indicates that autophagy can contribute to resistance against chemotherapeutics [88]. In recent years, it has become evident that autophagy might be critical in GIST progression. Beclin1 (BECN1) forms the BECN1–PIK3C3–PIK3R4 complex by aggregating cofactors, at the same time the autophagy protein cascade is activated [89]. Miselli and colleagues demonstrated that GIST patients responded to imatinib treatment by autophagy instead of apoptosis [90]. High levels of pro-autophagy beclin1/PI3K III and low levels of anti-autophagy beclin1/bcl2 complexes are consistent with the existence of autophagy in imatinib-treated GIST patients; thus suggesting autophagy playing a role in the underlying molecular mechanism of how GISTs form [90]. Wei and colleagues reported that beclin-1 knockdown significantly enhanced the sensitivity of GIST cells to imatinib; while miR-30a, directly targeting *Beclin1* also enhanced imatinib sensitivity through the downregulation of *Beclin1* [91]. Upon treatment, the sizeable GIST cell subpopulations survive and remain quiescent for a long time, leading to acquired resistance and treatment failure. Gupta and colleagues confirmed that a considerable number of GIST cells enter a reversible resting state by activating autophagy-dependent survival mechanisms after imatinib treatment [92]. GIST cells were destroyed by the synergistic effect of imatinib and autophagy inhibition by RNAi-mediated silencing against ATG7 and ATG12 [73]. The standard first-line molecular-targeted therapy in GISTs is focused on imatinib for GISTs; but primary or secondary resistance is becoming increasingly prominent. Fortunately, some studies have focused on new molecular mechanisms, such as autophagy, in order to address drug resistance. Hsueh and colleagues reported that NVP-AUY922, an HSP90AA1 inhibitor, of which KIT is a client, regulated autophagy-mediated pathways to downregulate the expression of the KIT protein and to inhibit GIST cell growth [93]. Rapamycin, an mTOR inhibitor and autophagy inducer, enhanced autophagy activity, downregulated the expression of KIT, and led to apoptosis in GIST430 and GIST48 cell lines [94]. 

## 4. Targeting the PI3K/AKT/mTOR Pathway via GIST Therapeutics

Most advanced GIST patients achieved remarkable clinical benefits from imatinib, but secondary resistance to imatinib was found in half of them [95]. The PI3K/AKT/mTOR pathway is suspended by inhibitors targeting different parts of the pathway [76]. Several inhibitors are currently in preclinical and early-stage clinical trials for GISTs (Figure 2 and Table 3).

### 4.1. Pure PI3K Inhibitors

PI3K inhibitors are subdivided into pan-inhibitors of class IA PI3Ks and isoform specific inhibitors [96]. Buparlisib (BKM 120) is a pan-PI3K inhibitor targeting all four isoforms of class I PI3Ks [97]. The in vivo study by Looy and colleagues supports the usefulness of this PI3K inhibitor in patient-derived GIST xenograft models [98]. They showed that BKM120 caused moderate tumor volume reduction in both imatinib-resistant xenograft models and inhibited the proliferative activity in the GIST48 xenograft model with imatinib-resistance caused by secondary *KIT^exon17^* mutation. BYL719 is a specific inhibitor that blocks the p110α catalytic domain of PI3K. Looy and colleagues demonstrated that BYL719 induced tumor volume reduction in the imatinib-sensitive UZLX-GIST3 xenograft model [98]. GDC-0941 is another pan-PI3K inhibitor. An in vivo study by Floris and colleagues showed the treatment strategy for GDC-094 plus imatinib combination as being very effective in several GIST xenograft models [99]. GDC-094 plus imatinib was more efficient than the single drugs in decreasing tumor burden and increasing anti-proliferative activity in all GIST xenografts [99].

Currently, for phase I clinical trials, imatinib and sunitinib refractory GIST patients are being recruited, and they will be treated with imatinib in combination with BKM120 (NCT01468688) and BYL719 (NCT01735968). These studies are expected to determine the maximum tolerated dose of BKM120 or BYL719 combined with imatinib for refractory GISTs.

### 4.2. AKT Inhibitors 

AKT inhibitors specifically inhibit the activation of AKT, block the activation of mTORC1, and control the downstream effects of the PI3K signaling cascade. The preclinical and clinical studies of AKT inhibitors in GISTs are limited. Perifosine is an allosteric inhibitor that targets the PH domain of AKT and prevents its translocation to the plasma membrane required for activation [100]. In a phase I/II study that investigated perifosine plus imatinib for imatinib-resistant metastatic GIST patients, most patients exhibited different degrees of toxicity reactions, manifesting as ocular toxicity and ulcerative keratitis [101].

### 4.3. mTOR Inhibitors

Rapamycin specifically binds to the FKBP12 binding protein that interacts with the mTORC1 complex, and then inhibits downstream signaling. Rapamycin’s analogs have the same effect [100]. Two rapamycin analogs used in clinic are everolimus (RAD001) and temsirolimus. RAD001 specifically targets mTOR and affects downstream signals. The in vitro study by Chang and colleagues demonstrated that, when the GIST82 cell line was treated with imatinib and RAD001 separately, the expression of phosphor-S6K1 that promotes cell growth and cell cycle progression was completely inhibited [102]. The authors did not describe the expression of phosphor-S6K1 when the GIST882 cell line received RAD001 plus imatinib as a combination. According to this study, RAD001 has effects on suppressing cell proliferation [102]. In vivo as single drugs, both RAD001 and imatinib displayed an anti-tumor effect in the GIST882 mouse model, but RAD001 was superior. Compared to single drug treatment, RAD001 plus imatinib was shown to significantly enhance tumor response to the drug in the GIST882 mouse model, indicating that the combination of RAD001 and imatinib is effective [103]. As a limitation, the authors did not assess PI3K and AKT activity levels upon drug application [103]. Most GIST patients benefit from imatinib, yet some will eventually exhibit resistance. Bauer and colleagues demonstrated an additive effect in the imatinib-sensitive GIST882 cell line for RAD001 plus imatinib, but not in imatinib-resistant GIST430 and GIST48 cell lines [75]. In the same study, upon treatment with 1μM imatinib, residual KIT activation rates of GIST430 and GIST48 cell lines were 6- and 2.8-fold higher than the GIST882 cell line, respectively. Therefore, secondary *KIT* mutations may be related to KIT hyperactivation and imatinib resistance [75].

In subsequent clinical studies, a phase I/II study (NCT01275222) evaluated the safety and efficacy of RAD001 plus imatinib in imatinib-resistant GIST patients, indicating that RAD001 plus imatinib was well tolerated by all patients [104]. In phase I, imatinib-resistant GIST patients were suggested to receive the optimal dose with imatinib (600/800 mg/day) combined with weekly (20 mg) or daily (2.5/5.0 mg) RAD001. In phase II, advanced imatinib-resistant GIST patients benefited from the optimal dose with imatinib 600 mg/day combined with RAD001 2.5 mg/day. The combination of rapamycin analogs and imatinib merits further investigation in imatinib-resistant GISTs. 

### 4.4. Dual PI3K/mTOR Inhibitors

The effect of mTORC1 inhibitors may be affected due to S6K1 and mTORC2 feedback loops, in which S6k1 inhibits mTORC1 and affects the activation of PI3K and AKT [100]. Thereby, dual PI3K and mTORC1/mTORC2 inhibitors, targeting all four isoforms of class I PI3K and mTORC1/mTORC2 complexes, are worth further studies and tests. PTEN, as a negative regulator of the PI3K-AKT pathway [105], is regulated mainly by single allele-deletion, which was significantly increased in imatinib-resistant GIST patients compared with imatinib-sensitive GISTs patients (39% vs. 9%) [106]. In the imatinib-resistant GIST430 cell line, imatinib combined with the dual PI3K/mTOR inhibitor dactolisib (BEZ235) induced the hyperphosphorylation of MAPK and only partially counteracted the activation of S6K. The overactivation of AKT and MAPK in both GIST cell lines with mono-allelic PTEN loss demonstrated that PTEN dysfunction caused the upregulation of PI3K/AKT and MAPK signaling [106]. In another study, BEZ235 plus imatinib remarkably improved the efficacy, with decreasing tumor volume and increasing pro-apoptotic effects in both imatinib-resistant (UZLX-GIST2) and imatinib-sensitive (UZLX-GIST3 and UZLX-GIST4) xenograft models [98].

Other inhibitors in clinical development, such as SF1126 and GDC-0980, were considered powerful for refractory GISTs. SF1126 is a small molecule pro-drug, which can convert to LY294002 [100]. In a phase I study of 39 patients with B-cell malignancies and advanced solid tumors, one GIST patient resistant to temsirolimus was treated with SF1126, and his disease was stable for more than one year, which may indicate that SF1126 had an effect on mTORC1 [107]. In another phase I study, three GIST patients treated with GDC-980 showed greater than 25% reduction in FDG uptake as assessed by PET. One of these patients showed 58% reduction in FDG uptake, indicating that GDC-980 has anti-tumor activity [108].

## 5. eIFs and PI3K/AKT/mTOR Pathway

### 5.1. The Relationship between eIFs and the PI3K/AKT/mTOR Pathway

Protein translation has committed effects on the process of eukaryotic gene expression [109,110]. The translation includes the following four steps: initiation, elongation, termination, recycling. It is well established that the rate-limiting step is the initiation phase regulated by eukaryotic translation initiation factors (eIFs) [109,111]. The initiation step starts with the assembly of an elongation-appropriate complex, of a 43S pre-initiation complex that is composed of the 40S small ribosomal subunit, methionine tRNAi, and a group of eIFs. Subsequently, the 43S pre-initiation complex binds to the 5′ end of mRNA and then to the eIF4F complex that mediates the recruitment of ribosomes to Mrna [111,112]. The eIF4F complex consists of three polypeptides (eIF4E, eIF4A and eIF4G). eIF4E is a cap-binding protein that recognizes and binds to the mRNA 5′ m^7^G cap structure, which interacts with the DEAD box RNA helicase eIF4A and the scaffolding protein eIF4G. The function of eIF4G is to bridge the mRNA with the ribosome, enhance the helicase activity of eIF4A, and realize the mRNA circularization [113]. Through the PI3K/AKT/mTOR pathway, extracellular stimuli induce phosphorylation and the inactivation of the 4Ebinding proteins (4E-BP1, 2 and 3) that influence eIF4F activity [112]. Thus, the high influence of eIFs on protein translation is of utmost interest in targeted cancer therapies. 

In a review about the coverage of the mTOR pathway by Next-Generation Sequencing (NGS) oncology panels, eIFs, a group of mTOR downstream proteins, were reported with low mutational frequencies of a rate less than 1% and no reported drug sensitivity alterations [114]. Numerous marker genes are known to have great impact on disease progression and prognosis, even though they are rarely mutated. eIF subunits are important examples of these markers [114].

### 5.2. PI3K/AKT/mTOR-Regulated eIFs as a Potential Therapeutic Target in Tumors 

eIFs may become promising therapeutic targets for many tumors under the PI3K/AKT/mTOR pathway control. The eIF4F complex is a critical part under the regulation of this signaling axis [113,115,116]. In Abl-expressing leukemic cells, eIF4B stimulated eIF4F activity by increasing the eIF4A RNA helicase activity on the mRNA 5′UTR, integrating signals from the PI3K/AKT/mTOR pathway [117]. When those inhibitors of PI3K (with LY294002), AKT (with Akti-1/2) or mTOR (with rapamycin) were separately applied to Abl-transformed cells, the phosphorylation of eIF4B Ser422 markedly decreased, which inhibited cell proliferation and growth [117]. Cencic and colleagues highlighted that eIF4A reversed drug resistance by curtailing the potential activity of translation initiation in lymphomas [118]. Hippuristanol, a translation initiation inhibitor, specifically inhibits eIF4A. Hippuristanol plus ABT-737 (an inhibitor of *Bcl-2)* synergistically increased cell death in Myc-driven lymphomas; likewise, the inhibition of eIF4AI is enough to improve the chemosensitivity of Myc-driven lymphomas to ABT-737 [118]. 

Furthermore, some studies focused on the PI3K/AKT/mTOR signaling members and other eIFs in human tumors. Golob-Schwarzl and colleagues demonstrated that eIF5 has the potential as a biomarker to indicate whether there is virus infection in hepatocellular carcinoma (HCC) tissue [119]. eIF5 was downregulated in non-virus related HCC; nevertheless, it revealed an overexpression in HBV-associated HCC. The downregulation of p-mTOR and mTOR was observed in HBV-associated HCC. The expression of other eIFs, such as eIF2a, eIF3D, eIF3H, eIF3I, eIF3J, eIF4E and eIF6 was downregulated in HBV-associated HCC [119]. Tapia and colleagues investigated the activation of PI3K/AKT/mTOR signals in gastric cancer (GC), with the overexpression of most important target proteins of the pathway, such as PI3K, AKT, p-AKT, p-mTOR, p-4E-BP1, P70S6K1, p-P70S6K1, eIF-4E, and p-eIF-4E proteins in tumor tissue [120]. Low expression of 4E-BP1 has a poor overall survival, which points towards a potential role as prognostic marker [120]. Wang and colleagues demonstrated EIF3B expression as upregulated in advanced GC patients who have a short 5-year survival [121]. In vitro and in vivo studies into the downregulation of EIF3B showed the inhibition cell proliferation and clonogenicity in SGC7901 and BGC823 cells lines, and knockdown of EIF3B notably abated the tumor volume and weight in an SGC7901 xenograft mouse model. Interestingly, the PI3K/AKT/mTOR signaling activity is related with an upregulation of eIF3B [121]. Golob-Schwarzl and colleagues showed the expression of eIFs family members, bringing eIF1, eIF5, and eIF6 to attention, together with components of the PI3K/AKT/mTOR signaling cascade in colorectal cancer (CRC) [122]. These eIF subunits and the PI3K/AKT/mTOR signaling members had a significant influence on the overall survival of CRC patients [122].

In summary, the PI3K/AKT/mTOR pathway has important effects on the basic intracellular functions, including cell growth, apoptosis, translation, and cell metabolism [44]. Especially, the regulation of translation is critical for maintaining homeostasis in cells [109]. The dysregulation of protein synthesis may be related to the abnormal expression of eIFs and altered activation of the PI3K/AKT/mTOR pathway [112,114]. 

## 6. Challenges and Future Directions

More than 80% of patients with advanced GISTs have remarkable clinical benefits from molecular targeted therapy, but secondary resistance is found in half of these GIST patients [37]. Imatinib as the first-line therapy effectively provides long-term disease stability and prolongs the overall survival in *KIT*-mutated GIST patients [37,123]. In a multicenter clinical study, only 10% of imatinib-treated patients had a 10-year progression-free survival, and interestingly GIST patients with *KIT exon 11* mutations suggest a better outcome beyond 10 years [124]. GIST patients with *KIT exon 9* mutations had a significantly worse prognosis compared to those with *KIT exon 11* mutations [123], who also exhibited worse response to imatinib treatment [125]. Other TKIs, such as sunitinib and regorafenib, have complementary activity in that sunitinib targets the ATP-binding pocket to inhibit imatinib-resistance mutations, whereas regorafenib primarily targets the activation loop to achieve similar effects [126]. Once imatinib resistance emerges, sunitinib and regorafenib as second- and third-line molecular targeted therapies, could provide clinical benefit in a short-term [127,128,129]. For imatinib-resistant GIST patients, the tumorigenesis mainly involved in the heterogeneity of *KIT* secondary mutations [130], and targeting *KIT,* is limited by its susceptibility to multiple mutations [125]. It may be a key to inverse secondary resistance in GISTs through selecting agents with known pre-clinical activity for mutations that form the basis for resistance, or that overcome the effect of resistance mutations. 

The PI3K/AKT/mTOR signaling axis is a crucial survival pathway in imatinib-resistant GISTs [131,132]. Subsequently, many studies paid attention on directly inhibiting downstream of this pathway in GISTs, in order to interrupt more comprehensive RTK signals as much as possible. Therefore, more research will have to pay attention to the potential of those inhibitors of PI3K (BKM120, BYL719, GDC-0941), AKT (Perifosine), mTOR (RAD001) and Dual PI3K/mTOR (BEZ2335, GDC-0980), combined with imatinib. Parts of combination treatment strategies are being investigated in early-stage clinical studies. For instance, BKM120 plus imatinib in combination could inhibit cell growth in imatinib-resistant GIST cell lines, whereas imatinb combined with GDC-0941 or with BEZ2335 improved the efficacy with a more pronounced tumor volume reduction in all GIST xenograft models. In addition, mTOR activity is related to the mutation type status. The mTOR pathway is activated in *PDGFRA* mutated and wt GISTs, which suggests that mTOR or upstream mTOR inhibitors may be potential therapeutic targets for primary resistant GISTs [79]. In mouse models, pharmacologic inhibitors of the PI3K pathway diminished tumor proliferation in tumor-bearing single-mutated *KIT^V558Δ/+^* mice, indicating that PI3K kinase makes a major contribution to tumor cell proliferation in established GISTs [133]. Thus, the interruption of the PI3K/AKT/mTOR pathway may represent a rational therapeutic approach in GIST patients.

Interestingly, the PI3K/AKT/mTOR pathway plays a pivotal role in mRNA translation, especially in the initiation phase. mTOR regulates the assembly of eIF4f to manage translation initiation. The PI3K/AKT/mTOR pathway affects tumorigenesis through eIFs, reported in HCC [119], GC [121], CRC [122], and other cancer types [117,134]. In summary, eIFs integrate the signals from the PI3K/AKT/mTOR pathway, which may be a promising target for tumor therapy in future. Regrettably, there is no literature on eIFs involved in GISTs. Based on the above principles, the hypothesis, that eIFs may represent a therapeutic target in GISTs via the PI3K/AKT/mTOR pathway, merits further investigation. 

## 7. Conclusions

Although most patients with advanced GISTs display remarkable clinical benefits from TKI therapy by imatinib, resistance to TKIs occurs in half of those GIST patients. Further therapeutic approaches are urgently needed. The PI3K/AKT/mTOR pathway is not only vital in tumorigenesis, but also important for novel potential therapeutic targets. To date, several inhibitors targeting different parts of the PI3K/AKT/mTOR pathway have been investigated in imatinib-resistant GISTs, including PI3K inhibitors, AKT inhibitors, mTOR inhibitors and dual PI3K/mTOR inhibitors. These drugs combined with imatinib may potentially provide a broader inhibition of pro-growth cellular mechanisms than the single-agent imatinib. Besides, the PI3K/AKT/mTOR pathway has a pivotal effect on mRNA translation initiation, directly regulated by eIFs that may also be a promising future therapeutic target for these tumors.

## Figures and Tables

**Figure 1 cancers-12-02972-f001:**
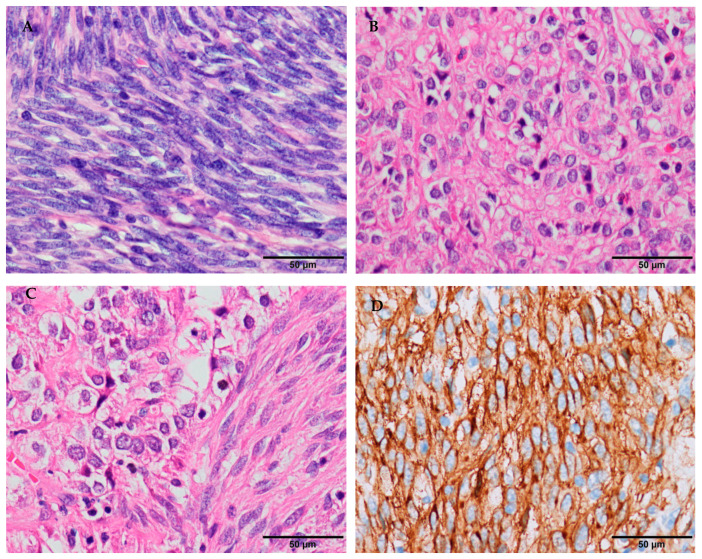
Histomorphology and immunohistochemistry in GISTs (own images, the scale bar is 50 μm). (**A**) Spindle cell type is mainly composed of fusiform cells with a spiral or palisading arrangement (H&E stain, ×400). (**B**) Epithelioid cell type is mainly composed of cords or small nests of epithelioid cells (H&E stain, ×400). (**C**) Mixed, spindle-epithelioid type is composed of spindly and epithelioid cells with a staggered arrangement (H&E stain, ×400). (**D**) Tumor cells are strongly positive for CD117 (EnVision stain, ×400). (**E**) Tumor cells are positive for DOG-1 (EnVision stain, ×400).

**Figure 2 cancers-12-02972-f002:**
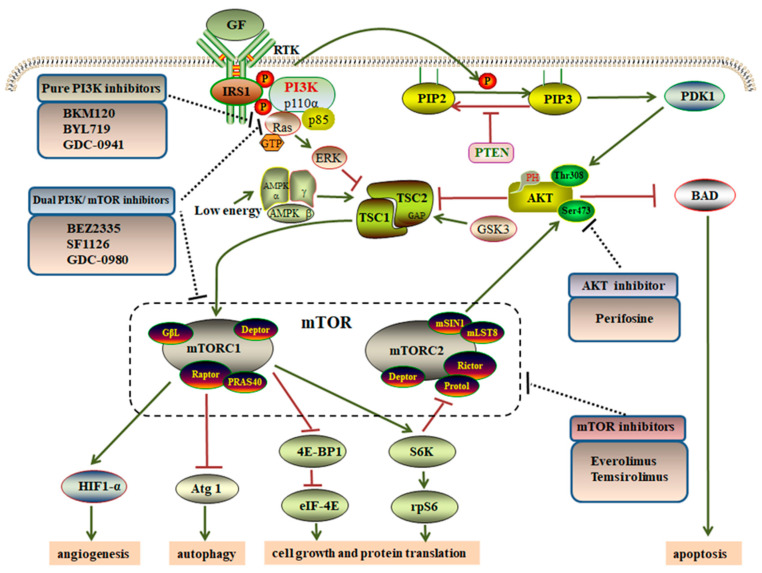
A schematic overview of the PI3K/AKT/mTOR pathway and inhibitors targeting different parts of the pathway in GISTs. Activation of RTKs by external growth factors leads to activate the PI3K/AKT/mTOR pathway, which directly and indirectly results in tumorigenesis, the inhibition of apoptosis and autophagy, the activation of protein translation, as well as angiogenesis. Inhibitors of PI3K, AKT, mTOR or dual PI3K/mTOR are being developed to treat refractory GISTs more effectively, by suppressing tumor progression. Abbreviations: mTORC: mammalian target of rapamycin complex; RTK: receptor tyrosine kinase; rpS6: ribosomal protein S6; IRS1: insulin receptor substrate 1; 4E-BP1: 4E-binding protein 1; AMPK: AMP-activated protein kinase; eIF4E: eukaryotic translation initiation factor 4E; ERK: extracellular signal-related kinase; GF: growth factors; HIF1-α: Hypoxia-inducible factor 1-α; PDK1: pyruvate dehydrogenase lipoamide kinase isozyme 1; S6K: ribosomal S6 kinase; AKT: protein kinase B; PI3K: phosphatidylinositol 3-kinase; PTEN: phosphatase and tensin homolog; TSC: tuberous sclerosis protein; PIP2: phosphatidylinositol 4,5-bisphosphate; ATG1: autophagy-specific gene 1 kinase; PIP3: phosphatidylinositol 3,4,5-trisphosphate; BAD: Bcl2-related death protein.

**Table 1 cancers-12-02972-t001:** Summary of pathological types and immunophenotypes in GISTs.

Feature	Percentage
Pathological type	
Spindle cell type	70%
Epithelioid cell type	17%
Mixed, spindle-epithelioid type	13%
Immunophenotype	
CD117-positivity	More than 90%
DOG1-positivity	96%

**Table 2 cancers-12-02972-t002:** Summary of genotypes in GISTs.

Genotype	Percentage
*KIT* mutation	
*KIT* exon 11 mutation	90%
*KIT* exon 9 mutation	8%
*KIT* exon 13 mutation	1%
*KIT* exon 17 mutation	1%
*PDGFRA*	5–8%
No *KIT* or *PDGFRA* mutation (including *SDH*-deficiency and non-*SDH* deficient)	10–15%

**Table 3 cancers-12-02972-t003:** Clinical trials in GISTs with therapeutic targets in the PI3K/AKT/mTOR pathway.

Drug	Trial Phase	Target	Condition	Status	Trial Number
BYL719+ Imatinib	Phase Ib	c-KIT + PI3K p110α	Third-line GISTs	Recruiting	NCT01735968
BKM120+Imatinib	Phase Ib	c-KIT + PI3K Class I	Third-line GISTs	Recruiting	NCT01468488
RAD001+ Imatinib	Phase II	c-KIT + mTOR	Progressive GISTs	ongoing	NCT00510354
RAD001+ Imatinib	Phase I/II	c-KIT + mTOR	Resistant GISTs	Completed	NCT01275222
Temsirolimus	Phase II	mTOR	GISTs	Completed	NCT00087074
Perifosine+ Imatinib	Phase II	c-KIT + AKT	Resistant GISTs	Completed	NCT00455559
Perifosine+ Sunitinib	Phase I/II	c-KIT + AKT	Advanced GISTs	Completed	NCT00399052

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
