# Peer review of "Therapeutic Potential of PI3K/AKT/mTOR Pathway in Gastrointestinal Stromal Tumors: Rationale and Progress"

_cancers, 2020, doi:10.3390/cancers12102972_

Round 1

Reviewer 1 Report

In the present review Duan Yi and colleagues have tried to elucidate the state of art of PI3K/AKT/mTOR pathway as potential therapeutic target in gastrointestinal stromal tumor (GIST). Overall, the paper is well written and clear in the study rational and procedures. Please find below few issues/comments to be addressed:

-Clinical feature section: Please provide few details regarding the epidemiology of GIST.

-Figure 1: Improve the quality of images A-B-C. Moreover, please replace figure D and E with 2 more suitable images representing positive CD117 and DOG-1 tumor samples, respectively.

-You never mention Figure 2 in the text. Please insert it.

-There are a couple of mistakes in the text. Please correct them.

Author Response

Response to Reviewer 1 Comments

Point 1: Clinical feature section: Please provide few details regarding the epidemiology of GIST.

Response 1: Thanks for your suggestions. We have provided the epidemiology details of GIST in line 46 to 50 as follows: “Søreide and colleagues reviewed 29 studies of more than 13,550 GIST patients from 19 countries. The median age was mid 60s across most studies. The incidence rate was 10 to 15 per million per year. Hong Kong, Shanghai, Taiwan and Norway (Northern part) were the highest incidence areas, with up to 19–22 per million per year; notably, Shanxi province of China was the area with the lowest incidence, with 4.3 per million per year”.

Point 2: Figure 1: Improve the quality of images A-B-C. Moreover, please replace figure D and E with 2 more suitable images representing positive CD117 and DOG-1 tumor samples, respectively.

Response 2: Thanks for your comment. The original images were indeed not clear enough. In order to highlight the morphological feature in a better quality, we replaced all images in Figure 1 with more appropriate new ones. The images are inserted between line 112 and 113.

Point 3: You never mention Figure 2 in the text. Please insert it.

Response 3: Thanks for your comment. We are sorry and kindly apologize for this mistake. Figure 2 has been inserted in line 196, 205, 213, 221, 239 and 318.

Point 4: There are a couple of mistakes in the text. Please correct them.

Response 4: We appreciate your comment. We did our best to correct typographical and grammatical errors and present now a clear and unambiguous paper.

Reviewer 2 Report

The manuscript written by Duan et al provides a detailed review of the features, characteristics, dysfunctional pathways, and therapeutic options of Gastrointestinal stromal tumor. While the review seems to be informative however, it suffers from the following facts which could consider as minor points:

1) In some parts the details get very long and descriptive. For instance, the pathology part could be summarized in one paragraph.

2) If the histo-images are taken from another study, the original paper should be cited. If not, then is better to be indicated that is self-prepared images. By the way, the magnifications and labeling inside the images are not clear.

3) It would be advantageous if in fig.1 D and E a better and higher magnification could be shown that readers could see which compartment of the cell is stained.

4) It would be more informative if the available inhibitors for each pathway or molecules shown in fig 2 could be displayed directly on the image.

Author Response

Response to Reviewer 2 Comments

Point 1: In some parts the details get very long and descriptive. For instance, the pathology part could be summarized in one paragraph.

Response 1: Thanks for your suggestions. In order to make our manuscript more concise, we made the following adjustments:

(1). The pathology parts are now composed of one paragraph only;

(2). Some inappropriate sentences have been modified or deleted, as follows:

----The line 80 to 82: “The cut surface of the tumor is usually tan-pink, yellowish brown to dark grayish yellow, and gray-white, often with focal hemorrhage and cystic degeneration” has been deleted .

----The line 102 to 110: “Other immunohistochemical analyses are also used for diagnosing GISTs: CD34 was a useful antibody in GISTs diagnosis before the CD117 antibody became available, and 70-80% of GISTs are positive for CD34. The sensitivity of CD34 is lower than that of CD117 and DOG1.[18, 21] 30-40% of GISTs are positive for smooth musclae actin (SMA), indicating the poor smooth muscle differentiation in GISTs.[14, 18] Positivity for S-100 protein has been implicated in neuronal differentiation, commonly occurring in melanocytes, glial cells and Schwann cells,[22] with only 5% positivity in GISTs. The diagnosis of GISTs is primarily based on clinical, morphological and immunohistochemical characteristics. In the majority of cases, the suspicion of GISTs has been confirmed by the combination of CD117 and DOG1 immunostaining” has been deleted.

----The line 153 to 154: “c-KIT mutation has been confirmed to be of prognostic significance for patients’ survival” has been deleted.

----The line 158 to 159: “PDGFRA-D824V mutation presents insert biological behavior and patients with this mutation are primarily resistant to imatinib” has been modified to “Patients with a PDGFRA-D824V mutation are primarily resistant to imatinib”.

----The line 270:”benign GISTs, potentially malignant GISTs and malignant” has been deleted.

----The line 312 to 313:”mTOR is a downstream intermediate of the PI3K/AKT signaling pathway; moreover, rapamycin can eliminate its inhibitory effects to enhance autophagy in GISTs” has been deleted.

----The line 344 to 345: “which seemed unlikely to treat refractory GISTs effectively by AKT inhibition. Up to now, clinical trial using this intervention strategy is not underway” has been deleted.

----The line 429 to 430: “the inhibition of eIF4AI urge ABT-737 more sensitive to murine lymphomas.” has been modified to “the inhibition of eIF4AI is enough to improve the chemosensitivity of Myc-driven lymphomas to ABT-737”.

Point 2: If the histo-images are taken from another study, the original paper should be cited. If not, then is better to be indicated that is self-prepared images. By the way, the magnifications and labeling inside the images are not clear.

Response 2: Thanks for your comment. The images are made by the authors. “own images” has been indicated in Figure 1. And the magnifications and labeling inside have been improved.

Point 3: It would be advantageous if in fig.1 D and E a better and higher magnification could be shown that readers could see which compartment of the cell is stained.

Response 3: Thanks for your point. We have improved the quality of Figure. 1 D and E as new pictures were taken. Now the immunohistochemistry in the images should be clear. The figures show that the cytoplasm and cell membrane are stained.

Point 4: It would be more informative if the available inhibitors for each pathway or molecules shown in fig 2 could be displayed directly on the image.

Response 4: This is an excellent idea. The available inhibitors for each pathway have been integrated into Figure 2 in our manuscript. So, Figure 3 was deleted.

Reviewer 3 Report

The authors describe the known facts about the participation of the PI3K / AKT / mTOR pathway in the development of GIST, including its role as a therapeutic target in these types of cancer. The authors also suggest the possibility of further exploring this pathway and its elements in targeted therapies.

The authors have collected a lot of data on this topic, so the publication is rich in cited data from the literature. Sometimes it is difficult to explore the content of this work, especially in the fragments that are based on enumerating the processes or factors involved in these processes. For this reason, I suggest that the authors prepare more tables based on this collected data, which will certainly be much more attractive to the reader. For example, the diagnostic methods used in GIST can be summarized in an interesting table. The different types of GIST can also be summarized tabularly.

The authors present microscopic photos showing the types of cells or various antigens stained on the sections. I understand that these are the authors' own data. Therefore, the legend of this figure should be richer. For example, what is the value of the scale bars, the data on the antibodies used, or the methodology generally used, would also be useful.

The authors did not avoid typing errors or mental shortcuts or other errors in the text. I am giving only examples below, and the author must carefully read the manuscript and correct it as a whole, or submit it for linguistic proofreading.

Here are some examples:

-line 369 "rapalogues" - is this an official abbreviation for rapamycin analogs?

-line 406 no spaces between words (this repeats itself in other places as well);

-line 434 - typing error

-line 439 a sentence starting with "In vitro and in vivo studies ..." - something is missing here.

Besides, after the authors carefully corrected and improved the text, this article should arouse the interest of readers.

Author Response

Response to Reviewer 3 Comments

Point 1: The authors have collected a lot of data on this topic, so the publication is rich in cited data from the literature. Sometimes it is difficult to explore the content of this work, especially in the fragments that are based on enumerating the processes or factors involved in these processes. For this reason, I suggest that the authors prepare more tables based on this collected data, which will certainly be much more attractive to the reader. For example, the diagnostic methods used in GIST can be summarized in an interesting table. The different types of GIST can also be summarized tabularly.

Response 1: Thanks for your suggestions. We have provided two tables to summarize the collected data of GIST in our manuscript.

----Table 1 is a summary of pathological types and immunophenotypes in GISTs between line 87 and 88.

----Table 2 is a summary of genotypes in GISTs between line 137 and 138.

Point 2: Therefore, the legend of this figure should be richer. For example, what is the value of the scale bars, the data on the antibodies used, or the methodology generally used, would also be useful.

Response 2: Thanks for your suggestion. The legends for the images have been up-dated in our manuscript, including the value of the scale bars and the methodology.

----For Figure 1. A, B and C, the images were stained with H&E and had a magnification of 400x.

----For Figure 1. E and F, the images were stained with Envision and had a magnification of 400x.

Point 3: The authors did not avoid typing errors or mental shortcuts or other errors in the text. I am giving only examples below, and the author must carefully read the manuscript and correct it as a whole, or submit it for linguistic proofreading. Here are some examples:-line 369 "rapalogues" - is this an official abbreviation for rapamycin analogs?-line 406 no spaces between words (this repeats itself in other places as well);-line 434 - typing error-line 439 a sentence starting with "In vitro and in vivo studies ..." -something is missing here.

Response 3: We appreciate the comment on rapalogs because the most established mTOR inhibitors are so-called rapalogs (rapamycin and its analogs). In order to prevent any misunderstanding we now call them rapamycin analogs in our manuscript when we mean rapamycin analogs. And we did our best to correct typographical or grammatical errors and present a clear and unambiguous paper.

Round 2

Reviewer 3 Report

This interesting review has been corrected by the authors. It is consistent, interesting, and worth publishing.